# IKKε Inhibitor Amlexanox Promotes Olaparib Sensitivity through the C/EBP-β-Mediated Transcription of Rad51 in Castrate-Resistant Prostate Cancer

**DOI:** 10.3390/cancers14153684

**Published:** 2022-07-28

**Authors:** Sophie Gilbert, Benjamin Péant, Anne-Marie Mes-Masson, Fred Saad

**Affiliations:** 1Centre de Recherche du Centre Hospitalier de l’Université de Montréal (CRCHUM) et Institut du Cancer de Montréal, Montréal, QC H2X 0A9, Canada; sophiegilbe@gmail.com (S.G.); peantb@yahoo.fr (B.P.); fredsaad@videotron.ca (F.S.); 2Department of Medicine, Université de Montréal, Montréal, QC H3C 3J7, Canada; 3Department of Surgery, Université de Montréal, Montréal, QC H3C 3J7, Canada

**Keywords:** combination therapy, DNA damage, DNA damage repair, transcription, PARP inhibitors

## Abstract

**Simple Summary:**

Most men with advanced hormone-sensitive prostate cancer (HSPC) treated with androgen deprivation therapy will develop castrate resistant prostate cancer (CRPC), a lethal form of prostate cancer (PC). Our group has previously shown that IKKε expression is stronger in CRPC tumors and correlates with aggressive PC. Moreover, we have shown that IKKε depletion or inhibition (BX795, Amlexanox) decrease CRPC cell proliferation and tumor volume in an in vivo mouse model. We also demonstrate that IKKε inhibitors specifically target CRPC to induce a senescent phenotype as well as DNA damage and genomic instability. In this study, we demonstrated that IKKε depletion or inhibition block C/EBP-β recruitment on Rad51 promoter to decrease promoter activity. We have also shown that Amlexanox treatment sensitizes CRPC cells to Olaparib in vitro and in mouse models. Taken together, targeting IKKε with Amlexanox combined with Olaparib may lead to additional effective therapeutic strategies in the management of patients with CRPC.

**Abstract:**

The progression of prostate cancer (PC) is often characterized by the development of castrate-resistant PC (CRPC). Patients with CRPC are treated with a variety of agents including new generation hormonal therapies or chemotherapy. However, as the cancer develops more resistance mechanisms, these drugs eventually become less effective and finding new therapeutic approaches is critical to improving patient outcomes. Previously, we have shown that IKKε depletion and IKKε inhibitors, BX795 and Amlexanox, decrease CRPC cell proliferation in vitro and in vivo and that IKKε inhibitors induce a senescence phenotype accompanied by increased DNA damage and genomic instability in CRPC cells. Here, we describe a new role for IKKε in DNA damage repair involving Rad51 and examine the therapeutic potential of Amlexanox combined with the PARP inhibitor Olaparib in CRPC cell lines. Combining Amlexanox with Olaparib decreased CRPC cell proliferation and enhanced DNA damage through the inhibition of Olaparib-induced Rad51 recruitment and expression in CRPC cells or IKKε-depleted PC-3 cells. We demonstrated that Rad51 promoter activity, measured by luciferase assay, was decreased with Amlexanox treatment or IKKε depletion and that Amlexanox treatment decreased the occupancy of transcription factor C/EBP-β on the Rad51 promoter. Our mouse model also showed that Amlexanox combined with Olaparib inhibited tumor growth of CRPC xenografts. Our study highlights a new role for IKKε in DNA damage repair through the regulation of Rad51 transcription and provides a rationale for the combination of Amlexanox and Olaparib in the treatment of patients with CRPC.

## 1. Introduction

Prostate cancer (PC) remains the most frequently diagnosed cancer in North American men. After radiation therapy or surgery, a quarter of patients will experience biochemical relapse and undergo androgen deprivation therapy for hormone-sensitive tumors. However, in 90% of cases, tumors will progress to castrate-resistant PC (CRPC) for which effective treatments are limited. Therefore, new therapeutic strategies to target CRPC and improve patient outcomes are needed.

Poly (ADP-ribose) polymerase 1 inhibitors (PARPi) such as Olaparib have been used as treatment for different cancers with defects in DNA damage repair. PARP1 is a member of the PARP family of enzymes required for the repair of DNA single-strand breaks (SSB), particularly during base excision repair (BER). Olaparib causes arrest of the replication fork, resulting in DNA double-strand breaks (DSB) that are normally repaired by homologous recombination (HR). However, Olaparib induces a synthetic lethality in tumor cells that carry mutations in DNA damage repair genes including *BRCA1*, *BRCA2*, *ATM*, and *RAD51* [1,2]. Almost 19% of patients with CRPC carry these mutations, which contribute to PC development and progression [3], and treatments with Olaparib alone or in combination with abiraterone or enzalutamide have been shown to improve the median overall survival in patients with metastatic CRPC [4,5,6].

IκB kinase-epsilon (IKKε), a member of the IKK family, is involved in the interferon (IFN) response through the phosphorylation of IFN regulatory factor (IRFs), IRF3 and IRF7 [7]. In prostate, breast and ovarian cancers, IKKε is overexpressed and involved in cancer progression [8,9,10]. In CRPC tumors, IKKε expression is dependent on NF-kB activity [11]. Péant et al., showed that IKKε overexpression leads to the phosphorylation and nuclear translocation of C/EBP-β, a transcription factor regulating IL-6 gene transcription, which contributes to inflammation, proliferation, progression to CRPC and development of bone metastases [12,13,14]. Importantly, IKKε is involved in protecting cells from DNA-damage-induced cell death through its translocation to the nucleus and formation of PML nuclear bodies [15]. This pro-survival function of IKKε in the nucleus of CRPC cells is dependent on the phosphorylation of IKKε itself and its targets, including the transcription factor C/EBP-β, which regulates the transcription of multiple genes involved in DNA damage repair [15,16]. In addition, IKKε depletion in CRPC cells decreases cell proliferation and tumor volume in a mouse model [13]. We have previously shown that IKKε inhibitors, BX795 and Amlexanox (an FDA-approved drug), also decreased CRPC cell proliferation and induced a senescence phenotype accompanied with DNA damage and genomic instability. Moreover, we demonstrated that BX795 and Amlexanox delay tumor growth in CRPC xenografts [17]. Taken together, these observations suggest that IKKε represents a promising target for the development of new treatments against CRPC.

In this study, we elucidated the role of IKKε in DNA damage repair and revealed how IKKε inhibition promotes DNA damage. Specifically, we examined IKKε in HR-mediated DNA damage repair and evaluated the combination of Amlexanox and Olaparib as a potential treatment to promote synthetic lethality in CRPC cell lines. We show that Amlexanox combined with Olaparib decreased CRPC cell proliferation and enhanced DNA damage. Amlexanox inhibited Olaparib-induced Rad51 recruitment by inhibiting C/EBP-β recruitment on the Rad51 promoter, thereby decreasing Rad51 transcription necessary for DNA repair. We also show that the Amlexanox-Olaparib combination more effectively delayed tumor growth compared to either treatment alone in a CRPC xenograft model.

## 2. Materials and Methods

### 2.1. Cell Lines and Cell Culture

Human PC cell lines used are PC-3 (CVCL_0035), DU145 (CVCL_0105), LNCaP (CVCL_0395) and 22Rv1 (CVCL_1045), which were purchased from the American Type Culture Collection, and C4-2B (CVCL_4784) cell line provided by Dr. Martin Gleave (Vancouver Prostate Center, BC, Canada). All cell lines were grown in RPMI 1640 (Wisent Inc., St-Bruno, QC, Canada) with 10% fetal bovine serum (FBS), 100 µg/mL gentamicin, and 0.25 µg/mL amphotericin B (Invitrogen, Paisley, UK). All PC cell lines were validated by short tandem repeat DNA profiling by the McGill University Genome Center (Montreal).

### 2.2. Cloning, Viruses and Infections

IKKε-depleted PC-3 cells were produced in previous study [17]. The lentiviral short hairpin RNAs (shRNAs) against IKKε and RFP were purchased from Dharmacon (Chicago, IL, USA). Infected cells were selected with puromycin (1 µg/mL) and cells were either used immediately or harvested, frozen and stored at −80 °C.

### 2.3. Drugs

Amlexanox was purchased from Abcam (Cambridge, UK), and Olaparib was obtained from SelleckChem. Drugs, initially dissolved in 100% DMSO, were diluted in RPMI medium and added to cells 24 h after cell seeding.

### 2.4. Cell Proliferation Assays

PC cells were seeded at 3000 cells/well in 96-well plates. After 24 h, cells were treated with different concentrations of Amlexanox and/or Olaparib and incubated for 6 days. Cell proliferation was analyzed using an IncuCyte Zoom Live-Cell Imaging System (IncuCyte HD, Essen BioScience, Ann Arbor, MI, USA). Cell confluence values were determined using Incucyte Zoom Software with frames captured at 2-h intervals.

### 2.5. Drug Combination Analysis

The combination drug effect was evaluated using CompuSyn software ComboSyn Inc., Paramus, NJ, USA, as indicated by Fleury et al. [18]. CRPC cells were treated with 50, 100, 150 or 200 μM Amlexanox and/or 0.25, 0.5, 1 or 2 μM Olaparib for PC-3 and C4-2B cells or 2.5, 5, 10 or 20 μM for DU145 cells. Cell confluence values were determined with Incucyte Zoom Software and Bliss score values were calculated to define synergestic (positive values), additive (value of zero) and antogonisme (negative value).

### 2.6. Western Blot Analysis

Proteins were extracted from PC cell lines using lysis buffer (Triton X-100, 50 mM Tris pH 7.4, 150 mM NaCl, 2 mM EDTA and 10% glycerol) containing a protease inhibitor cocktail (Roche Applied Science, Indianapolis, IN, USA) and phosphatase inhibitors (5 mM NaF, 200 µM Na_3_VO_4_ and 100 µM PMSF) and quantified by Bradford assay. Thirty micrograms of protein were separated with precast 4–15% Tris-Glycine SDS-polyacrylamide gels (MINI-PROTEAN TGX Gels, BioRad Laboratories, Hercules, CA, USA) and transferred onto PVDF membranes with the Trans-Blot Turbo Transfer System (BioRad). Membranes were blocked for 1 h in TBS-Tween supplemented with 5% of milk then probed with primary antibodies overnight at 4 °C. Membranes were washed with TBS-Tween three times for 10 min and probed with peroxidase-conjugated secondary antibodies (Millipore, Temecula, CA, USA) for 1 h. Chemiluminescence was enhanced by ECL prime (Amersham, GE Healthcare, Chicago, IL, USA) and detected using the ChemiDoc MP Imaging System (Bio-Rad). The antibodies used in this study were anti-IKKε (1/1000, Imgenex, San Diego, CA, USA), anti-Rad51 (1/1000, Abcam) and anti-ß-actin (1/2000, Abcam).

### 2.7. Total RNA Extraction, Reverse Transcription and qPCR

Before 24 h, cells were seeded at density of 400,000 cells/well in 6-well plates. Cells were treated with 100 µM Amlexanox and/or 10 µM Olaparib and incubated for 48 h. Cells were trypsinized and pelleted. Total RNA extraction was performed using the RNeasy Mini Kit (Qiagen, Hilden, Germany), according to the manufacturer’s protocol, and quantified by NanoDrop (Thermo Fisher, Waltham, MA, USA). For reverse transcription reaction, one microgram of RNA was used and performed using Quantitect Reverse Transcription kit (Qiagen) with the following conditions: 2 min at 42 °C, 15 min at 42 °C, and 3 min at 95 °C. One microliter of reverse transcription products were incubated with specific primers (forward and reverse, 400 nM) and the SYBR Select Master Mix (Applied Biosystems). The reference gene used is the TATA-box binding protein (TBP). Primers included the following:

TBP-Fw 5′-GAGCCAAGAGTGAAGAACAG 3′;

TBP-Rev 5′ ACCTTATAGGAAACTTCACATCAC 3′;

Rad51-Fw 5′ CAACCCATTTCACGGTTAGAGC 3′;

Rad51-Rev 5′ TTCTTTGGCGCATAGGCAACA 3′. 

Reactions were performed in the Applied 63 BioSystems^®^ Step One Plus system (UDG activation 50 °C/2 min, followed by AmpliTaq activation plus denaturation cycle 95 °C/2 min and 40 cycles at 95 °C/15 s, 60 °C/1 min and 72 °C/30 s). Target gene expression was normalized to TATA-box binding protein (TBP) levels. For each sample, the fold change value was expressed as:

2^−ΔΔCT^: ΔΔCT = (CT_target gene_ − CT_TBP_)control − (CT_target gene_ − CT_TBP_)treated. 

### 2.8. Immunofluorescence

Cells were seeded at 60,000 cells onto coverslips in 24-well plates. After 24 h, cells were treated with 100 µM Amlexanox and/or 10 µM Olaparib for 48 h. After washing with PBS, cells were fixed with formalin 10% (Chaptec) for 10 min then washed again with PBS. For γH2AX foci detection, cells were permeabilized in PBS supplemented with 0.25% Triton for 20 min and, after washing, were blocked with PBS supplemented with 1% bovine serum albumin (BSA) and 4% donkey serum for 30 min. For Rad51 foci detection, cells were permeabilized in PBS supplemented with 0.5% Triton and 0.2 N HCl and blocked with PBS supplemented with 8% BSA and 4% FBS for 30 min. Cells were then incubated with primary anti-γH2AX antibody (Millipore) or anti-Geminin (GMNN) (Proteintech Group, Rosemont, Il, USA) and anti-Rad51 overnight at 4 °C. After three washings with PBS, cells were incubated with Alexa Fluor^®^ Cy5 goat anti-mouse IgG secondary antibody or Alexa Fluor^®^ 488 goat anti-rabbit IgG secondary antibody (Life Technologies Inc., Carlsbad, CA, USA) for 1 h. Coverslips were washed with PBS and water and then mounted on the slides using ProLong™ Gold Anti-fade Mountant with DAPI (Life Technologies Inc.). Zeiss AxioObserver Z1 fluorescent microscope (Carl Zeiss, Jena, Germany) was used to acquire images (mosaics 4 × 4). The quantification was done with ImageJ software to count and calculate the average number of foci per nucleus.

### 2.9. EdU (5-ethynyl-2′-deoxyuridine) Detection

Cells were seeded at density of 60,000 cells/well onto coverslips in 24-well plates. After 24 h incubation, PC cells were treated with 100 µM Amlexanox and/or 0.5, 10 or 0.25 µM Olaparib, for PC3, DU145 and C4-2B cells, respectively for 24 h. Then, 10 μM EdU (Invitrogen) was added for 24 h at 37 °C without change the media. Cells were washed with PBS and fixed with formalin for 10 min. To detect EdU incorporation, cells were incubated with staining solution (100 mM Tris pH 8.5, Vitamin C, CuSO_4_ and Alexa Fluor 647 dye) for a 30-min in the dark. After washing, coverslips were mounted onto slides with ProLong™ Gold Antifade Mountant with DAPI. Zeiss AxioObserver Z1 fluorescent microscope (Carl Zeiss, Jena, Germany) was used to acquire images (mosaics 4 × 4). To count the total cell number and positive cells for EdU staining, images were analyzed with ImageJ software.

### 2.10. Colony Formation

Cells were seeded at 1000 cells/well in 6-well plates and incubated. After 24 h, cells were treated with 100 µM Amlexanox and/or 0.5, 10 or 0.25 µM Olaparib, for PC3, DU145 and C4-2B cells, respectively µM for 6 days. Cells were fixed with methanol and stained with 50% *v*/*v* methanol and 0.5% *w*/*v* methylene blue solution (Sigma-Aldrich, Saint-Louis, MO, USA). Colonies were counted using a stereomicroscopy. Each experiment was repeated three times.

### 2.11. Luciferase Assay

Cells were seeded at 400,000 cells per well in 6-well plates and incubated for 24 h. CRPC cells or IKKε-depleted PC-3 cells were transfected with Rad51 promoter luciferase vector (SwitchGear Genomics, Active-motif, Carlsbad, CA, USA) using lipofectamine 2000 reagent according to the manufacturer’s protocol. CRPC cells were then treated with 100 µM Amlexanox for 48 h and then collected with trypsin, centrifugated and resuspended in PBS. An aliquot was lysed and proteins were quantified by Bradford assay. Luciferase activity was measured on the rest of the sample using the LightSwith Luciferase assay kit (SwitchGear Genomics) according to the manufacturer’s protocol. Luciferase activity was normalized using protein quantification for each condition. Each experiment was repeated three times.

### 2.12. Chromatin Immunoprecipitation

Cells were seeded in 150 mm Petri dishes and transfected with pORF9-C/EBP-β-flag2 vector (derived from pORF-C/EBP-β (Invivogen) [13]) using lipofectamine 2000 reagent. Cells were treated with 100 µM Amlexanox or untreated as control. After 48 h, chromatin was prepared using the SimpleChIP Enzymatic Chromatin IP kit (magnetic beads) (#9003S, Cell signaling Technology, Danvers, MA, USA) according to the manufacturer’s protocol. Briefly, cell proteins were crosslinked to DNA with 1% formaldehyde for 10 min at 37 °C and then incubated with 1× Glycine for 5 min at room temperature. After washing with PBS, cells were collected and centrifugated at 2000× *g* for 5 min. Cell pellets were resuspended with 1× Buffer A (supplemented with DTT and PIC) and incubated on ice for 10 min to lyse the cells. Nuclei were collected by centrifugation at 2000× *g* for 5 min and washed with 1× Buffer B (supplemented with DTT). Then, each pellet was resuspended with 1× Buffer B with micrococcal nuclease and incubated for 20 min at 37 °C to digest DNA. Digestion was stopped with 0.5 M EDTA followed by centrifugation at 16,000 *g* for 1 min. The nuclear pellet was resuspended with 1× ChIP Buffer (supplemented with PIC) and sonicated with four pulses (15 s) to generate DNA fragments. Lysate was clarified by centrifugation at 9400× *g* for 10 min. For each immunoprecipitation, 10 µg of chromatin was incubated with 2 µg of anti-Flag antibody (F1804, Sigma) overnight at 4 °C. The next day, 30 µL magnetic beads were added for 2 h at 4 °C. After three low salt washes and one high salt wash, chromatin was eluted at 65 °C for 30 min in 400 µL 1× ChIP Elution Buffer. After addition of 6 µL NaCl and 2 µL proteinase K, chromatin was incubated at 65 °C for 2 h to reverse crosslinking. DNA was purified using DNA purification columns provided in the kit. Quantitative PCR using POWER SYBR green master mix (Thermofisher Scientific, Waltham MA, USA) was performed on 2 µL of DNA with the Rad51 promoter primers: Fw 5′—CCCCCGGCATAAAGTTTGA—3′ and Rev 5—GCTTTCAGAATTCCCGCCA—3′. ChIP signal was normalized to input cross-linked DNA for each sample.

### 2.13. Murine Xenograft Model

All animal experiments were conducted with ethical regulations for animal testing and research at CRCHUM (Centre de recherche du Centre hospitalier de l’Université de Montréal) and approved by our institutional committee on animal care (Comité Institutionel de Protection des Animaux—CIPA—protocol number C14011AMMs). PC-3 and DU145 cells were prepared at 1 million cells/200 µL in PBS–matrigel (*v*/*v*) and subcutaneously injected into the flank of 8-week-old male NRG mice (NODRag1^null^, IL2rg^null^, NOD rag gamma, from our in-house mouse colony). Measurements of tumor size were followed twice a week and weight of mice once a week. To calculate the tumor volumes, the following equation was used: V (mm^3^) = a × b × h (a is the largest diameter, b is the perpendicular diameter and h is the height). Before drug injection, mice were randomized into four groups: control (carrier 1: PBS-5% DMSO and carrier 2: 40% PEG300, 10% DMSO, 0.72% NaCl), Amlexanox (25 mg/kg) in carrier 2, Olaparib (50 mg/kg) in carrier 1 and Amlexanox plus Olaparib in the appropriate carrier. Carriers, Amlexanox and Olaparib were administered by intraperitoneal injection every day for 3 weeks. 

### 2.14. Immunofluorescence on Xenograft Tissue 

Harvested tumors were fixed with formalin and embedded in paraffin. Tissue sections were cut with thick of four micrometers using a microtome. Antigen retrieval step and primary antibody staining were performed using the automated Ventana Discovery XT staining system (Ventana Medical Systems, Oro Valley, AZ, USA) according to the manufacturer’s instructions. Anti-GMNN antibodies were added on slides and incubated at 37 °C for 60 min. Then, blocking solution Dako (Agilent Technologies, Palo Alto, CA, USA) was incubated for 20 min followed by secondary antibodies step (anti-rabbit Cy5; Life Technologies Inc.) for 45 min. To quench tissue autofluorescence, slides were incubated for 15 min with Sudan Black (0.1% (*w*/*v* in 70% ethanol). Slides were then mounted with ProLong^®^ Gold Antifade Mountant with DAPI and stored at 4 °C. To acquire images, a Zeiss microscopy was used and quantification was performed using ImageJ software.

### 2.15. Statistical Analysis

Data were expressed as the mean ± SEM. To perform the statistical analyses, GraphPad Prism software was used and determined using either the Student *t*-test or two-way ANOVA. All in vitro experiments were repeated at least three times and the number of mice is indicated in Section 3 for each cohort.

## 3. Results

### 3.1. Effect of the Amlexanox-Olaparib Combination on CRPC Proliferation

We have previously shown that IKKε inhibitors BX795 and Amlexanox decrease CRPC proliferation through the induction of a senescence phenotype accompanied with DNA damage and genomic instability [17]. To characterize the role of IKKε in the HR-mediated DNA damage repair induced by Olaparib, we evaluated the Amlexanox-Olaparib combination in the CRPC cell lines PC-3, DU145 and C4-2B. Real-time imaging cell proliferation assays showed no effect of Olaparib-tested concentrations on CRPC cell growth after 4 days (Figure 1A). In PC-3 cells, 100 µM Amlexanox significantly decreased cell proliferation, whereas the combination of Olaparib (0.5 µM) and Amlexanox (100 µM) led to a stronger reduction (Figure 1A and Appendix A). In DU145 cells, Amlexanox alone had no effect, but the combination of 100 µM Amlexanox and 10 µM Olaparib significantly decreased cell proliferation (Figure 1A and Appendix A). For C4-2B cells, the antiproliferative effect of 0.25 µM Olaparib was further enhanced by 100 µM Amlexanox (Figure 1A and Appendix A). Interestingly, Bliss scores were determined to evaluate the drug combination effect. Bliss score results have shown that the combination has a synergic effect: 9.6, 22.4 and 10.9 scores for PC-3, DU145 and C4-2B cells, respectively (Figure 1A).

The Amlexanox-Olaparib combination was also tested in hormone-sensitive PC (HSPC) cell lines (Appendix A). Previously, we have shown that HSPC cells do not overexpress IKKε, and IKKε inhibitors have no effect on HSPC proliferation and DNA damage induction [17]. Similarly, the Amlexanox-Olaparib combination did not affect cell proliferation of LNCaP and 22Rv1 cells (Appendix A).

We confirmed these observations by clonogenic assays after 6 days of treatment (Figure 1B). In PC-3 cells, Amlexanox and Olaparib treatments decreased colony formation by 62% and 48%, respectively, and the combination resulted in a decrease of 87%. In DU145 cells, colony formation decreased by 30% and 64% with Amlexanox and Olaparib treatments, respectively, whereas the combination decreased the number of colonies by 96%. In C4-2B cells, 42% and 66% decreases in colony formation were observed with Amlexanox and Olaparib, respectively, and a 91% decrease was observed with the combination (Figure 1B). These results were confirmed by analysis of cell cycle progression using short EdU pulse-labelling assays after 48 h of treatment (Figure 1C,D). In PC-3 cells, Amlexanox or Olaparib decreased EdU incorporation by 42% and 6%, respectively, and by 70% with the combination. DU145 cells showed a decrease of 26%, 29% and 74% in DNA synthesis after Amlexanox, Olaparib or the combination, respectively. Finally, a decrease of 26%, 57% and 84% of EdU incorporation was observed after Amlexanox, Olaparib or the combination, respectively (Figure 1C,D) in C4-2B cells. These results showed that Amlexanox can enhance the efficiency of Olaparib treatment in CRPC cells.

### 3.2. Amlexanox-Olaparib Combination Increases DNA Damage

To investigate how Amlexanox enhanced Olaparib-induced DNA damage, the appearance of γH2AX foci, a marker of DNA damage, was monitored by immunofluorescence after 48 h of treatment (Figure 2). In PC-3 and C4-2B cells treated with the Amlexanox-Olaparib combination, γH2AX foci increased by 15- and 3-fold, respectively, compared to control and either treatment alone. In DU145 cells, Amlexanox treatment alone induced more DNA damage (6-fold) than olaparib (2-fold), and the combination induced even more damage (11-fold). These results indicate that the Amlexanox-Olaparib combination is associated in CRPC cell lines with increasing DNA damage.

### 3.3. Amlexanox Inhibits Olaparib-Induced Rad51 Recruitment

Rad51 is one of the major proteins involved in DSB repair mediated by the HR pathway [19,20,21]. The formation of Rad51 foci in CRPC cells after Olaparib treatment was followed using immunofluorescence. Rad51 foci increased by 3, 3.8 and 5.8-fold in PC-3, DU145 and C4-2B cells, respectively (Figure 3A–F). In contrast, Amlexanox treatment inhibited Rad51 foci formation in PC-3, DU145, and C4-2B cells (Figure 3A–F). To discern whether Rad51 recruitment was inhibited or if the DNA repair rate was increased, a Rad51 recruitment assay with irradiation, which has been well characterized [22,23], was performed. Irradiation (8 Gy) induced maximal Rad51 recruitment at 2 h and increased Rad51 foci in PC-3, DU145 and C4-2B cells (3.1, 3.6 and 4.3-fold, respectively; Figure 3D–F and Appendix A). Amlexanox treatment inhibited this irradiation-induced Rad51 recruitment in CRPC cell lines. 

We then examined Rad51 foci formation in HSPC cells in which IKKε expression is inducible (Appendix A). Rad51 foci were induced by Olaparib treatment in LNCaP and 22Rv1 cells. However, Amlexanox treatment did not inhibit Olaparib-induced Rad51 recruitment in these cells (Appendix A). To confirm that the inhibition of Olaparib-induced Rad51 recruitment was directly due to Amlexanox inhibition of IKKε expression, IKKε in PC-3 cells was depleted using lentiviral shRNAs (Figure 3D) and the Rad51 foci induction measured in IKKε-depleted PC-3 cells. Olaparib treatment induced Rad51 foci by 4-fold in PC-3 cells transfected with the control shRFP but did not increase Rad51 foci in IKKε-depleted cells (Figure 3E,F). These results show that IKKε is involved in Rad51 recruitment during DNA damage repair in CRPC cells and that IKKε inhibition by shRNA or Amlexanox impairs the Rad51 recruitment induced by Olaparib. 

### 3.4. Effect of Amlexanox on the Rad51 Expression

Amlexanox treatment or IKKε depletion increased the sensitivity of CRPC cells to Olaparib by inducing DNA damage and blocking Rad51 foci recruitment for DNA repair. To understand how IKKε inhibition blocked Rad51 foci recruitment, levels of the Rad51 mRNA and protein were measured. After 48 h, Amlexanox treatment significantly decreased the basal levels of Rad51 mRNA and protein in PC-3, DU145 and C4-2B cells (Figure 4A,B). We also followed the Rad51 mRNA levels in HSPC cell lines. Amlexanox treatment decreased basal levels of Rad51 mRNA in LNCaP cells but did not change basal levels in 22Rv1 cells (Appendix A). While Olaparib increased the Rad51 expression (mRNA and protein) in PC-3, DU145 and C4-2B cells, the combination with Amlexanox blocked Olaparib-induced Rad51 expression (Figure 4A,B). This effect was not observed in LNCaP and 22Rv1 cell lines (Appendix A). In IKKε-depleted PC-3 cells, IKKε depletion decreased the basal level of Rad51 expression (Figure 4C) and was similar to the Amlexanox effect on CRPC cells, blocking Olaparib-induced Rad51 expression (Figure 4A,B). These results suggest that IKKε is involved in the induction of Rad51 transcription in CRPC cells. 

### 3.5. Effect of Amlexanox on the Regulation of Rad51 Mediated by C/EBP-β

To confirm the role of IKKε in Rad51 expression, the promoter region of wild-type Rad51 was cloned into the pLightSwitch_Prom reporter vector with the RenSP luciferase gene. Amlexanox treatment significantly repressed luciferase activity with a 2.5-fold decrease in PC-3, DU145 and C4-2B cells compared to control (Figure 5A). Olaparib did not affect activity in PC-3 and C4-2B cells but showed decreased activity in DU145 cells. For all three cell lines, the Amlexanox-Olaparib combination significantly repressed luciferase activity. Luciferase activity in IKKε-depleted PC-3 cells (Figure 5B) was then assessed. While Olaparib treatment increased the promoter activity 2-fold in shRFP cells, IKKε depletion combined with or without Olaparib treatment significantly repressed activity. These results demonstrate the involvement of IKKε in Rad51 promoter activity and expression.

Since IKKε is involved in C/EBP-β activation, which induces IL-6 transcription in CRPC cells [13] and is also involved in DNA damage repair [24,25], we localized the consensus binding site for C/EBP-β in the Rad51 promoter using the bioinformatics software CiiiDER (http://www.ciiider.org, accessed on 1 January 2020) (Figure 5C). To confirm that C/EBP-β binds to the Rad51 promoter, chromatin immunoprecipitation assays with Amlexanox treatment in CRPC cells were conducted. To perform this experiment, CRPC cells were transfected with a vector expressing a flag-tagged C/EBP-β protein (Figure 5D). Amlexanox treatment significantly decreased C/EBP-β occupancy of Rad51 endogenous promoter region by 40%, 54% and 60% in PC-3, DU145 and C4-2B cells, respectively (Figure 5E). IKKε-depleted PC-3 cells were also transfected with a flag-tagged C/EBP-β (Figure 5F). IKKε depletion induced a significant decrease of C/EBP-β fixation in the Rad51 promoter region (Figure 5G). These results suggest that IKKε regulates Rad51 transcription through C/EBP-β.

We investigated the effects of the Amlexanox-Olaparib combination in a CRPC xenograft mouse model using PC-3 and DU145 cell lines. NRG mice bearing established PC-3 and DU145 tumor xenografts were treated daily with intraperitoneal injections of Amlexanox (25 mg/kg), Olaparib (50 mg/kg) or Amlexanox-Olaparib combination for 3 weeks. A combination of the carriers was used as control. Treatments, alone or in combination, were well tolerated by the mice with no change in body weight and behavior. PC-3 and DU145 tumor growth were significantly more affected by the combination than each treatment alone (Figure 6). PC-3 tumor growth was significantly decreased by 16% with Amlexanox treatment and was not affected by Olaparib treatment, whereas the Amlexanox-Olaparib combination significantly delayed PC-3 tumor growth by 34% (Figure 6A). DU145 tumor growth was significantly decreased by 20% and 29% with Amlexanox or Olaparib treatment, respectively, while the combination decreased DU145 tumor growth by 36% (Figure 6B).

### 3.6. Amlexanox-Olaparib Combination Impairs the Growth of CRPC Xenograft 

To further determine the proliferative rate in tumors treated by the combination or each treatment alone, harvested xenograft tissue was stained using anti-GMNN antibodies (Figure 6C–F). GMNN staining showed that Amlexanox or Olaparib treatment altered the proliferation rate in PC-3 xenografts with a decrease of 44% and 26%, respectively, while the combination induced a significant decrease of 66% (Figure 6C,E). Similarly, GMNN straining was decreased with Amlexanox and Olaparib by 43% and 33%, respectively, in DU145 xenografts and the combination resulted in a staining decrease of 70% (Figure 6D,F). These results indicate that the Amlexanox-Olaparib combination was more effective in inhibiting in vivo antitumor growth compared to either treatment alone.

## 4. Discussion

IKKε was first described as an oncogene in breast cancer [26,27] and is often overexpressed in a variety of cancers, including ovarian, lung, pancreatic and prostate cancers [10,12,16,28,29,30,31]. IKKε overexpression is also associated with poor outcomes in patients with gastric and pancreatic cancers [30,31,32]. The therapeutic potential of IKKε has been demonstrated through IKKε depletion, which decreased cell proliferation and tumor growth for breast, gastric, colorectal and prostate cancers [13,31,33,34]. Moreover, IKKε inhibitors, such as Amlexanox, BX795 or CYT387, show anti-proliferative activity in vitro and in vivo in breast cancer as well as CRPC cells [17,35]. Amlexanox is the only IKKε inhibitor approved by the FDA and is clinically used for the treatment of buccal ulcers and asthma. However, different preclinical studies have shown that Amlexanox has potential in the treatment of cancer, demonstrating antitumoral effects in melanoma, glioblastoma and prostate cancer [17,36,37,38]. In addition, favorable results were obtained when Amlexanox was combined with other therapies, such as docetaxel in breast cancer [39] or MEK inhibitors in Non-Small Cell Lung Cancer [40]. Here, we demonstrated that Amlexanox combined with Olaparib decreased CRPC cell proliferation through inhibition of DNA damage repair via the control of Rad51 gene expression. In contrast to CRPC cells, HSPC cells have inducible IKKε expression and are not sensitive to Amlexanox and BX795 [17]. In this study, we observed that the Amlexanox–Olaparib combination also had no effect on HSPC cell lines. With few therapeutic options to treat patients with CRPC, our results show that Amlexanox combined with Olaparib could have a potential role as a targeted therapy specific for CRPC in patients with functional HR.

Several studies have shown that IKKε is involved with DNA damage. After genotoxic stress, IKKε translocates into the nucleus to form a complex with PML nuclear bodies and protect cells from DNA damage-induced cell death [15]. However, treatment with the IKKε inhibitors, BX795 and Amlexanox, results in an increase of DNA damage associated with multinuclei and micronuclei structures [17]. Here, we have shown that Amlexanox as well as IKKε depletion in CRPC cells may contribute to this increased DNA damage by blocking Rad51 recruitment and expression, which are involved in HR-mediated DNA repair. More specifically, Amlexanox repressed the Rad51 promoter activity by decreasing the occupancy of the C/EBP-β transcription factor. A recent study showed that a C/EBP-β knockdown impacted HR activity through decreased expression of BRCA1, BRIT1 or Rad51 [25]. In addition, C/EBP-β enhances PARPi resistance by inducing HR in high-grade serous ovarian cancer [25]. In CRPC cells, C/EBP-β is phosphorylated and activated by IKKε to induce IL-6 transcription [13]. Our results suggest that the IKKε overexpression in CRPC cells may activate C/EBP-β to induce Rad51 transcription and allow repair of DNA damage. IKKε inhibition by Amlexanox or IKKε knockdown prevents the activation of C/EBP-β, blocking Rad51 transcription and the expression of DNA repair proteins, thereby enhancing sensitivity of CRPC cells to inducers of DNA damage, such as Olaparib.

Olaparib targets tumors with mutations in DNA damage repair genes, such as *BRCA* and *ATM*. In CRPC, these mutations are found in 20% of patients [3] who have an Olaparib response rate of 88% as opposed to 6% for patients who do not carry these mutations [41]. Due to these mutations, tumor cells are defective in HR DNA repair and are more sensitive to loss of BER repair induced by Olaparib, thus resulting in cell death by synthetic lethality. In this study, we showed that Amlexanox blocks the Rad51 recruitment induced by Olaparib and inhibits Rad51 transcription mediated by C/EBP-β, rendering cells more susceptible to DNA damage. 

We also have previously shown IKKε involvement in PC progression with increased expression of IKKε in a patients tumor being linked with an increase in the PC aggressiveness [12]. Hence, our results introduce another approach to Olaparib-induced synthetic lethality with the inhibition of IKKε to target CRPC cells.

## 5. Conclusions

Overall, our study highlights a new function of IKKε and a potential therapeutic avenue with the combination of Amlexanox and Olaparib to target CRPC. We show that IKKε depletion and Amlexanox inhibit Rad51 expression, resulting in enhanced DNA damage and decreased HR efficiency. Our findings support the combination of Amlexanox and Olaparib for clinical development for the treatment of CRPC.

## Figures and Tables

**Figure 1 cancers-14-03684-f001:**
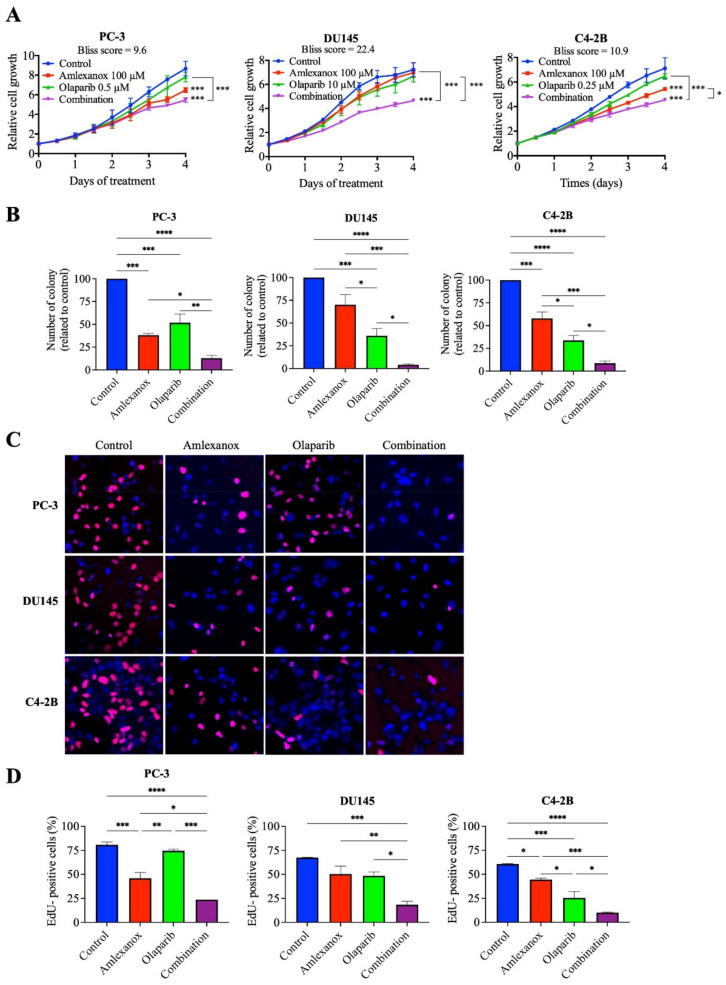
Antiproliferative effect of Amlexanox and Olaparib in CRPC cell lines. (**A**) Impact of Amlexanox and/or Olaparib on CRPC cell proliferation. Untreated or treated cells were incubated in 96-well plates and untreated (control) or treated with indicated concentrations of Amlexanox and/or Olaparib. Cell proliferation was followed by Incucyte live-cell imaging for 4 days. (**B**) Clonogenic assays were performed on CRPC cells treated with 100 µM Amlexanox and/or 0.5, 10 or 0.25 µM Olaparib, for PC3, DU145 and C4-2B cells, respectively, for 6 days. Cells were fixed and stained with methylene blue. Colonies were counted under a stereomicroscope. (**C**,**D**) Representative images (**C**) and quantification (**D**) of EdU-positive CRPC cells (red; blue for DAPI) after 24-h pulses of EdU and treatment with 100 µM Amlexanox and/or 0.5, 10 or 0.25 µM Olaparib, for PC3, DU145 and C4-2B cells, respectively. Each experiment was repeated three times. Statistical significance was determined by two-way ANOVA. * *p* < 0.05; ** *p* < 0.001; *** *p* < 0.0001; **** *p* < 0.00001.

**Figure 2 cancers-14-03684-f002:**
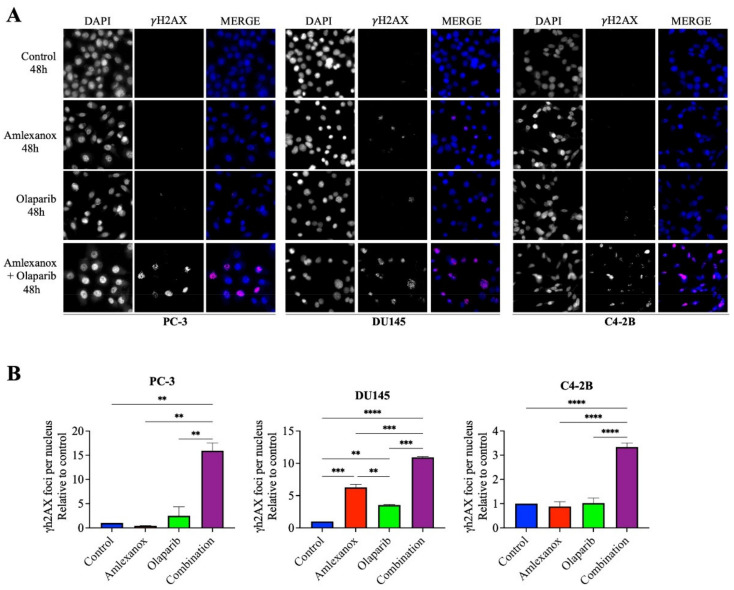
Amlexanox enhances Olaparib-induced DNA damage. Representative images (**A**) and quantification (**B**) of γH2AX foci in CRPC cells treated 100 µM Amlexanox and/or 10 µM Olaparib for 48 h. Cells were fixed and γH2AX foci (red) and DAPI (blue) were stained by immunofluorescence. Pictures were taken using a Zeiss microscope and the number of γH2AX foci per nuclei was counted using ImageJ. Each experiment was repeated three times. Statistical significance was determined by two-way ANOVA. ** *p* < 0.001; *** *p* < 0.0001; **** *p* < 0.00001.

**Figure 3 cancers-14-03684-f003:**
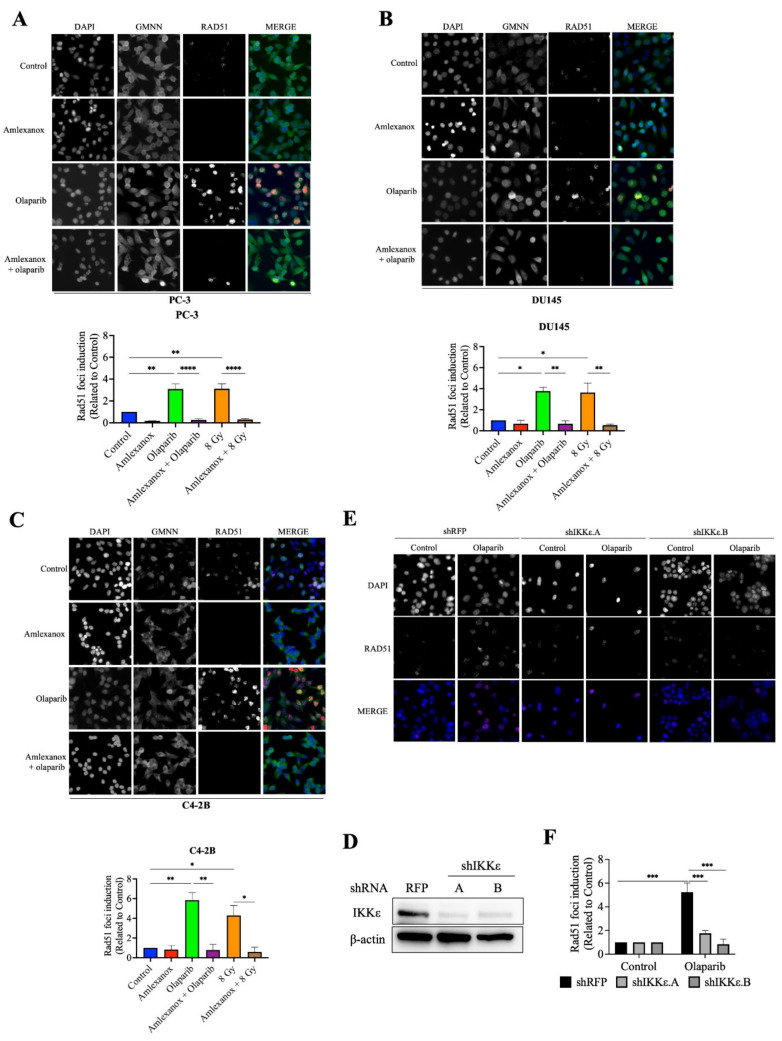
Amlexanox inhibits Olaparib-induced Rad51 recruitment. (**A**–**C**) Representative images and quantification of Rad51 foci in CRPC cells treated with 100 µM Amlexanox (48 h) and/or 10 µM Olaparib (for 24 h). Cells were fixed and stained for Rad51 foci (red), GMNN (green) and DAPI (blue). Pictures were taken using a Zeiss microscope and the number of Rad51 foci per GMNN-positive cells were counted using ImageJ. (**D**) PC-3 cells were transfected with lentiviral shIKKε.A or shIKKε.B. Western blots confirmed IKKε depletion with whole cell extracts separated, transferred and probed with IKKε antibody. (**E**) Representative images and (**F**) quantification of Rad51 foci in IKKε-depleted PC-3 cells. Each experiment was repeated three times. Statistical significance was determined by two-way ANOVA. * *p* < 0.05; ** *p* < 0.001; *** *p* < 0.0001; **** *p* < 0.00001. The full Western blots are shown in Appendix A.

**Figure 4 cancers-14-03684-f004:**
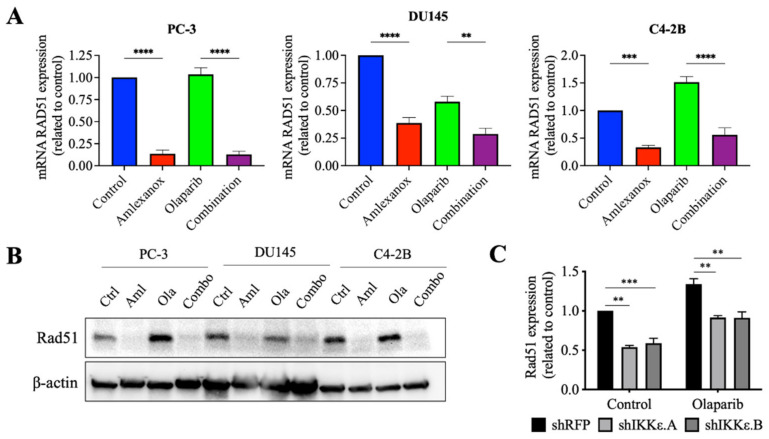
Amlexanox decreases Rad51 transcription and expression. (**A**,**B**) Rad51 expression in CRPC cells measured by RT-qPCR (**A**) and Western blot (**B**) after 48 h of 100 µM Amlexanox (Aml) and/or 10 µM Olaparib (Ola) treatment. (**C**) Rad51 expression in IKKε-depleted PC-3 cells treated with 10 µM Olaparib for 48 h and measured by RT-qPCR. Statistical significance was determined by two-way ANOVA. ** *p* < 0.001; *** *p* < 0.0001; **** *p* < 0.00001. The full Western blots are shown in Appendix A.

**Figure 5 cancers-14-03684-f005:**
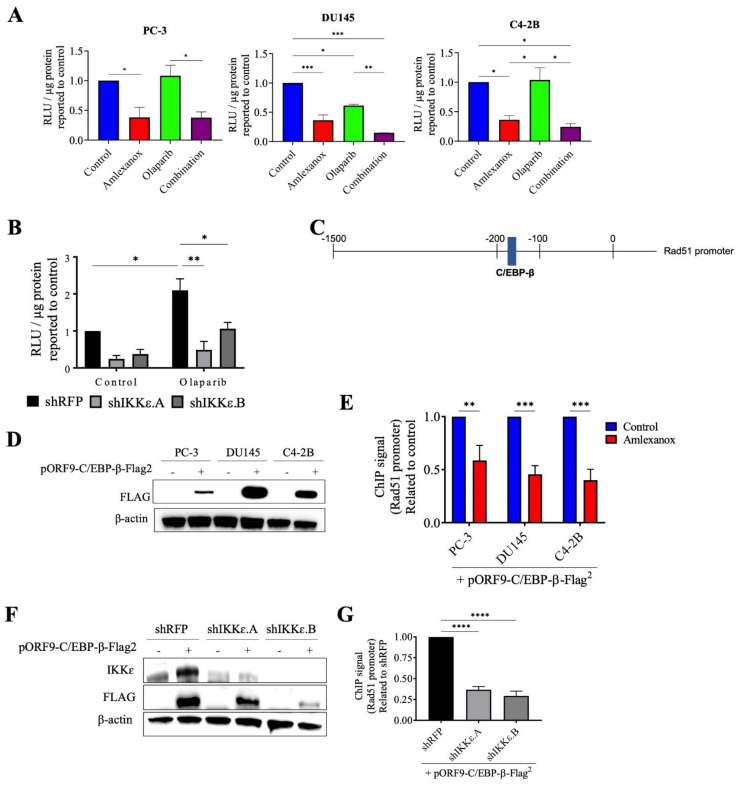
Amlexanox decreases Rad51 transcription by inhibiting C/EBP-β activation. (**A**,**B**) Rad51 promoter activity in CRPC cells (**A**) and in IKKε-depleted PC-3 cells (**B**) measured by relative light units (RLU) from luciferase assays. (**C**) Schematic representation of Rad51 promoter with C/EBP-β binding site. (**D**) CRPC cells were transfected with empty vector or pORF9-C/EBP-β-Flag2 vector. After 48 h, cells were collected and proteins analyzed by Western blot using anti-Flag antibody. (**E**) Chromatin immunoprecipitation (ChIP) was performed using anti-Flag antibody on chromatin extracts from CRPC cells that were transfected with pORF9-C/EBP-β-Flag^2^ vector and treated with 100 µM Amlexanox. (**F**) IKKε-depleted PC-3 cells were transfected with empty vector or pORF9-C/EBP-β-Flag2 vector. (**G**) Chromatin immunoprecipitation was performed using anti-Flag antibody on chromatin extracts from IKKε-depleted PC-3 cells transfected with pORF9-C/EBP-β-Flag^2^ vector or control. * *p* < 0.05; ** *p* < 0.01; *** *p* < 0.001; **** *p* < 0.0001. The full Western blot are shown in Appendix A.

**Figure 6 cancers-14-03684-f006:**
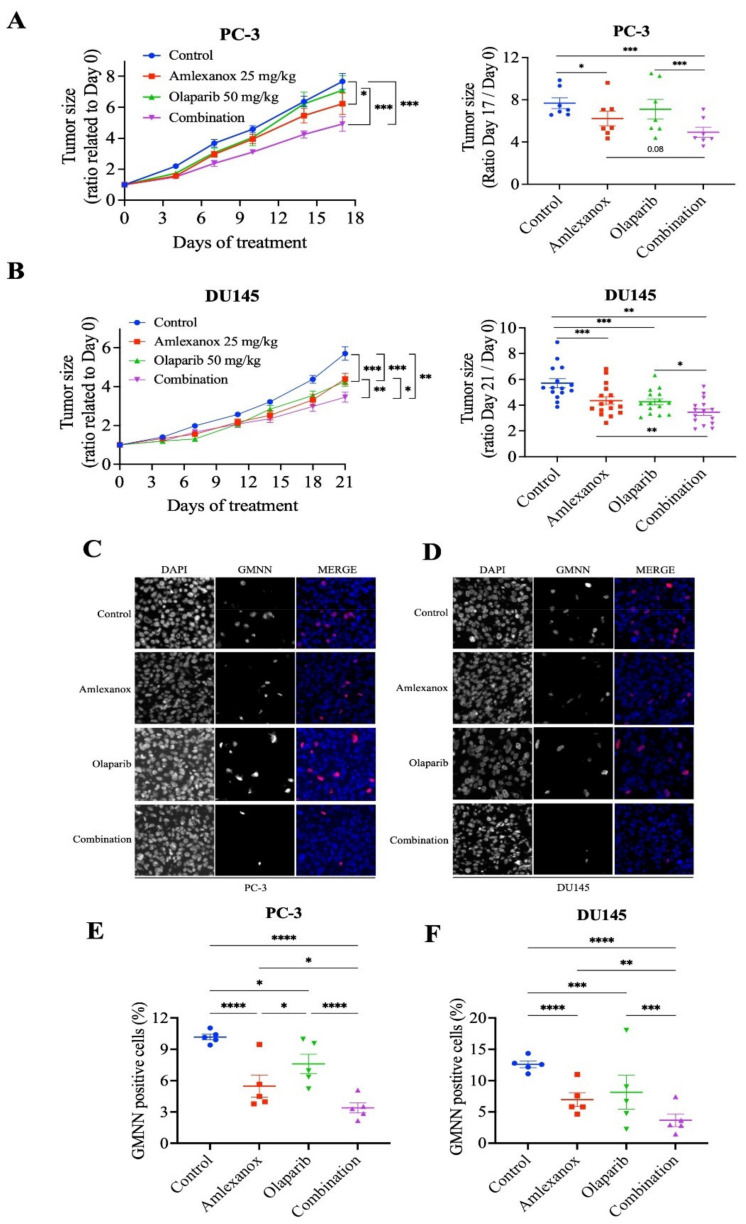
Amlexanox-Olaparib combination delays CRPC xenograft tumor growth in mouse model. (**A**,**B**) Tumor size ratio between the first and last days of treatment for PC-3 and DU145 xenograft tumors (*n* = 9 and *n* = 18, respectively). Amlexanox (25 mg/kg) and/or Olaparib (50 mg/kg) were administered daily by intraperitoneal injection. Tumor size was measured twice a week and all mice were sacrificed at endpoint. (**C**–**F**) Representative images (**C**,**D**) and quantification (**E**,**F**) of percentage of GMNN-positive cells (red) in PC-3 (**C**,**E**) and DU145 (**D**,**F**) xenografts (*n* = 5 per condition). Statistical significance was determined by two-way ANOVA. * *p* < 0.05; ** *p* < 0.01; *** *p* < 0.001; **** *p* < 0.0001.

## Data Availability

The data presented in this study is available in this article (and Appendix A).

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
