# Peer review of "IKKε Inhibitor Amlexanox Promotes Olaparib Sensitivity through the C/EBP-β-Mediated Transcription of Rad51 in Castrate-Resistant Prostate Cancer"

_cancers, 2022, doi:10.3390/cancers14153684_

Round 1

Reviewer 1 Report

IKKe Inhibition – Gilbert et al

Order of primers listing, p. 3-4, lines 144-147 – makes more sense to list Rad51 primers before the TBP primwers, especially since TBP is not defined until line 151.

Luciferase assay, p. 4, line 193 – More details should be provided on how cells are “collected and resuspended” e.g., assume cell are trypsinized and quenched?

“CRCHUM” on p.5, line 227 – this abbreviation should be identified/spelled out

Figure 1A, p. 7  – Some explanation of what Bliss score is, how numbers were interpreted, references related to the same,  should be provided either in figure legend or in manuscript text.

P. 8, line 308 - The phrase “To determine how Amlexanox enhance Olaparib-induced DNA damage…” seems inappropriate here.  Consider replacing “determine” with “investigate” or “confirm”, which more accurately conveys what the experimental data is accomplishing.

Figure 2B, p. 8 - **’s and ***’s are missing from all three graphs.

P.8, line 332 – a “fold-increase” in Rad51 foci value is missing for one of the 3 cell lines,

3.5-fold and 4-fold are indicated for two cell lines, but what is the third “fold-increase” value ?

Figure 3 & legend, p.9-10 – layout is confusing.  A & B labels are missing from figure. Also, it would be clearer if the first 3 micrograph arrays were all in one row and the corresponding bar graphs were each immediately below those.  Also suggest that the legend be included on the same page as the figure.

Figure 5A, p. 11 – text labeling on x-axis of the first 3 bar graphs is messy (irregular spacing and some character overlaps) and needs fixing.

Figure 5 legend – Suggest that entire figure and legend be moved onto the same page.

Section 3.6 – There seems to be some discrepancy between the percentage of inhibition by the combination Amlexonox-Olaparib treatment, between the tumor-size measurements and the proliferation-rate measurements (~35% decrease in tumor-size and ~65% decrease in proliferation-rate) – Why do the authors think this is?  Can they offer any explanation for the difference in tumor cell proliferation-rate inhibition and the in vivo tumor-size inhibition?

 Figure 6, p. 13 – Recommend changing layout so that bar graphs of E and F are immediately under the corresponding micrographs of C and D.

“BER repair” and general discussion of drug combinations p. 14, line 490-499 – this is the first mention of BER repair in the manuscript and 

The further discussion of therapeutic combination of Olaparib with androgen-receptor-signaling inhibitors, seems incongruous within this paragraph and with the main focus of this paper, which is IKKe inhibition.   Recommend that BER repair comment be either deleted or explained in more detail.  Also recommend removing the two sentences in line 491 to 496 regarding hormone agents, to keep the discussion more focused.

Author Response

Point 1 : Order of primers listing, p. 3-4, lines 144-147 – makes more sense to list Rad51 primers before the TBP primwers, especially since TBP is not defined until line 151.

Response 1: TBP has been defined before the primers sequences.

Point 2 : Luciferase assay, p. 4, line 193 – More details should be provided on how cells are “collected and resuspended” e.g., assume cell are trypsinized and quenched?

Response 2 : After 48 hours of treatment, cells were collected with trypsin. After centrifugation, cells were resuspended in PBS. The main text has been modified to provide these details (p5)

« CRPC cells were then treated with 100 µM Amlexanox for 48 hours and then collected with trypsin, centrifugated and resuspended in PBS.»

Point 3 : “CRCHUM” on p.5, line 227 – this abbreviation should be identified/spelled out

Response 3: This abbreviation was been spelled out. P5

« CRCHUM (Centre de recherche du Centre hospitalier de l’Université de Montréal)»

Point 4 : Figure 1A, p. 7  – Some explanation of what Bliss score is, how numbers were interpreted, references related to the same,  should be provided either in figure legend or in manuscript text.

Response 4 : Indeed, Bliss score results have not been described. A section in the Materials and Methods (p3) has been added to describe how bliss score results were obtained. In the results section (p6), the bliss scores are now described.

Point 5 : P. 8, line 308 - The phrase “To determine how Amlexanox enhance Olaparib-induced DNA damage…” seems inappropriate here.  Consider replacing “determine” with “investigate” or “confirm”, which more accurately conveys what the experimental data is accomplishing.

Response 5 : «Determine» was replaced by «investigate» (p9)

Point 6: Figure 2B, p. 8 - **’s and ***’s are missing from all three graphs.

Response 6 : This is a formatting  error that has now been rectified.

Point 7 : P.8, line 332 – a “fold-increase” in Rad51 foci value is missing for one of the 3 cell lines,

3.5-fold and 4-fold are indicated for two cell lines, but what is the third “fold-increase” value ?

Response 7 : the missing value has been added.

Point 8 : Figure 3 & legend, p.9-10 – layout is confusing.  A & B labels are missing from figure. Also, it would be clearer if the first 3 micrograph arrays were all in one row and the corresponding bar graphs were each immediately below those.  Also suggest that the legend be included on the same page as the figure.

Response 8: the layout of Figure 3 has been changed as requested.

Point 9 : Figure 5A, p. 11 – text labeling on x-axis of the first 3 bar graphs is messy (irregular spacing and some character overlaps) and needs fixing.

Response 9: Again this was a formatting error that has now been corrected.

Point 10 : Figure 5 legend – Suggest that entire figure and legend be moved onto the same page.

Response 10: the figure and legend are now on the same page

Point 11 : Section 3.6 – There seems to be some discrepancy between the percentage of inhibition by the combination Amlexonox-Olaparib treatment, between the tumor-size measurements and the proliferation-rate measurements (~35% decrease in tumor-size and ~65% decrease in proliferation-rate) – Why do the authors think this is?  Can they offer any explanation for the difference in tumor cell proliferation-rate inhibition and the in vivo tumor-size inhibition?

Response 11: It has been demonstrated that cell size is poorly correlated with cell death measurements. Indeed, it is now known that apoptotic cells are often larger than their normal counterparts. Therefore, it might be expected that tumors remain larger than anticipated even with a decrease in proliferation.

Point 12: Figure 6, p. 13 – Recommend changing layout so that bar graphs of E and F are immediately under the corresponding micrographs of C and D.

Response 12: The layout has been modified

Point 13: “BER repair” and general discussion of drug combinations p. 14, line 490-499 – this is the first mention of BER repair in the manuscript and 

The further discussion of therapeutic combination of Olaparib with androgen-receptor-signaling inhibitors, seems incongruous within this paragraph and with the main focus of this paper, which is IKKe inhibition.   Recommend that BER repair comment be either deleted or explained in more detail.  Also recommend removing the two sentences in line 491 to 496 regarding hormone agents, to keep the discussion more focused.

Response 13: We agree. Theses sentences have been deleted.

Reviewer 2 Report

Gilbert et al. reported that the combination therapy of Amlexanox and Olaparib impairs the proliferative potential of CRPC cell lines. The IKKε inhibitor, amlexanox reduces the olaparib promoted Rad51 recruitment which is important for DNA damage repair. The luciferase and ChiP assays revealed that binding of C/EBP-β onto the promoter region of Rad51 was attenuated by the treatment of amlexanox. Finally, the combination therapy inhibited the growth of CRPC xenografts in vivo. Overall this study very well designed and performed. However the following comments can further strengthen this manuscript.

1. In this study, authors used a single dose concentration of Amlexanox (100 µM) and different concentrations of Olaparib (0.5, 10 and 0.25 µM) for each cell lines. It is not clear that what would be the rationale in selecting these concentrations against tested cell lines. Is this based on calculated IC50 for each cell line from your data (If yes, include that data) or from published data (literature) and explain in the results section.

2. Combination treatment of Amlexanox and Olaparib in CRPC cells increases DNA damage. It would be important to show that increased DNA damage leads to apoptosis which would be evaluated by Annexin-FITC assay and represent the percentage of apoptotic cells with the treatments of drugs.

3. A moderate grammatical changes and typos needs to be corrected.

Author Response

  1. In this study, authors used a single dose concentration of Amlexanox (100 µM) and different concentrations of Olaparib (0.5, 10 and 0.25 µM) for each cell lines. It is not clear that what would be the rationale in selecting these concentrations against tested cell lines. Is this based on calculated IC50 for each cell line from your data (If yes, include that data) or from published data (literature) and explain in the results section.

Response: This value of a single dose concentration of Amlexanox is based on our previous article. This reference was quoted in first sentences of results section.

  1. Combination treatment of Amlexanox and Olaparib in CRPC cells increases DNA damage. It would be important to show that increased DNA damage leads to apoptosis which would be evaluated by Annexin-FITC assay and represent the percentage of apoptotic cells with the treatments of drugs.

Response: While this results on apoptosis might be of interest, the main focus of this experiment was to understand cellular changes that are induced by the combination strategy and as such address an important consequence of the combination.

  1. A moderate grammatical changes and typos needs to be corrected.

Response: The article has been reviewed for grammar.

Round 2

Reviewer 2 Report

I recommend this manuscript for publication.